# An analysis of neuroscience and psychiatry papers published from 2009 and 2019 outlines opportunities for increasing discovery of sex differences

Rebecca K. Rechlin[1,2,6], Tallinn F. L. Splinter[2,3,6], Travis E. Hodges[1,2], Arianne Y. Albert [2,4] & Liisa A. M. Galea [1,2,5✉]

Sex differences exist in many neurological and psychiatric diseases, but these have not always been addressed adequately in research. In order to address this, it is necessary to consider how sex is incorporated into the design (e.g. using a balanced design) and into the analyses (e.g. using sex as a covariate) in the published literature. We surveyed papers published in 2009 and 2019 across six journals in neuroscience and psychiatry. In this sample, we find a 30% increase in the percentage of papers reporting studies that included both sexes in 2019 compared with 2009. Despite this increase, in 2019 only 19% of papers in the sample reported using an optimal design for discovery of possible sex differences, and only 5% of the papers reported studies that analysed sex as a discovery variable. We conclude that progress to date has not been sufficient to address the importance of sex differences in research for discovery and therapeutic potential for neurological and psychiatric disease.

[1] Department of Psychology, University of British Columbia, Vancouver, BC, Canada. [2] Women's Health Research Cluster, University of British Columbia, Vancouver, BC, Canada. [3] Department of Biology, University of British Columbia, Vancouver, BC, Canada. [4] Women's Health Research Institute, BC Women's Hospital, Vancouver, BC, Canada. [5] Djavad Mowafaghian Centre for Brain Health, University of British Columbia, Vancouver, BC, Canada. [6] These authors contributed equally: Rebecca K. Rechlin, Tallinn F. L. Splinter. ✉email: liisa.galea@ubc.ca

The consideration of sex in published reports is essential to our understanding of disease and the biological mechanisms that contribute to the aetiology, manifestation and treatment of disease[1]. The study of sex differences is critical to our understanding of precision medicine in finding effective treatments for disease. Sex differences exist in the prevalence and manifestation of a number of neurological and psychiatric diseases[2,3]. Females are more likely to be diagnosed with multiple sclerosis, major depressive disorder, and have a greater lifetime risk of Alzheimer's Disease compared to males, whereas males are more likely to be diagnosed with autism spectrum disorder, attention and hyperactivity disorder, and Parkinson's Disease[1–4]. Even in diseases that do not show strong sex differences in prevalence, age of disease onset or manifestation can be different between the sexes[5,6]. There are notable differences in time to diagnosis[7], disease progression[2,4], vaccine response[8] and treatment efficacy/drug response[9] in a number of diseases. Harnessing the knowledge that males and females can differ on several disease-related outcomes will be fruitful in not only understanding disease but also in determining whether sex-specific risk factors for disease may warrant further attention. For example, sex differences in the manifestation of cardiovascular disease has prompted calls for changes to the diagnostic guidelines for cardiovascular disease based on sex[10]. To make headway for precision medicine and most effective treatment and diagnoses, sex must be taken into consideration in the design and analyses of data.

Many health disparities in treatment and diagnosis have been attributed to the lack of research in females in both animal models and in clinical work, and insufficient inclusion of women in clinical trials[11,12], and research funding agencies have attempted address this. For example, in the U.S. to increase the enrolment of women in clinical research, the United States Congress passed The Revitalisation Act of 1993. This Act stated that women and minorities must be included as subjects in clinical trials funded by the National Institutes of Health (NIH). However, implementation of the requirement of women and minorities has not translated into analysis by sex, race or ethnicity[13]. The importance of sex consideration in research led the NIH to further mandate the inclusion of women and racialized and ethnic minorities in clinical research in 2001, and the addition of sex as a biological variable (SABV) in biomedical research in 2016[14]. However, these requirements did not include specifications as to the analysis of the data by sex[15] nor did they specify sample size requirements[16]. Other countries have notable differences in their recommendations, timeline and mandates. The Canadian Institutes of Health Research (CIHR) implemented Sex and Gender-Based Analysis (SGBA) in 2010 as a mandatory component, and in 2019, into the scoring of grants. Horizon Europe (European Commission) has worked on policy changes since 2002 requiring the integration of sex and gender in research where relevant[17] and in 2020 Horizon Europe indicated the need for analyses of gender and sex (Supplementary Fig. 1). Although prescriptive guidelines from funding agencies are lacking there are a number of reviews with suggestions on the appropriate incorporation of SABV and SGBA in the literature[18–20]. Despite the mandates and recommendations from those funders, only a small improvement is observed with regards to their implementation[21–23]. This could potentially be due to reviewers and authors of papers applying SABV and SGBA inconsistently[24,25].

The biomedical and clinical research community is beginning to make corrections for a long-standing bias of using males predominately in research. Beery and Zucker[26] surveyed a sample of human and animal research papers across ten disciplines for sex inclusion in studies from a sample of papers in 2009 and although there was considerable variation by research field, the majority of papers sampled were not using both sexes[26]. Studies in human populations were more likely to use both males and females across the ten disciplines examined compared to studies using animals in their sample[26]. A 10-year follow up was done that demonstrated that there was an increase from 29% in 2009 to 49% in 2019 of papers that had studies that included both sexes, with neuroscience having one of the largest increases across disciplines between 2009 to 2019 in their sample[22]. Even though a greater proportion of papers are reporting studies that are including both sexes, there remain issues in how sex has been included. Approximately one-third of papers reporting sex-inclusive studies did not specify the sample size[22] and a large majority of papers reporting studies that used males and females failed to analyse the data by sex in 2009[22,26] Furthermore, between 2019 and 2009, there was an 8% decrease in the papers that used sex-based analyses[22] with only one discipline (pharmacology) showing an increase in the percentage of papers that used sex-based analyses from 2009 to 2019. Furthermore, a sex bias favouring the use of only male participants in papers is still prevalent in neuroscience research[22,27]. A study from Will and colleagues[27] indicated that the use of solely males in neuroscience papers they surveyed increased from 2010 to 2014 to ~40%, whereas the number of papers using solely females remained at a constant low value (5%). Thus, across the 10 years, research indicates that although the sex omission rate is decreasing across disciplines, the use of sex in the analyses and the large differential in single-sex studies favoring males has not appreciably changed[21,22].

What has been lacking in the literature is a detailed assessment of how sex is reported in papers (whether the study design is balanced and sex is used consistently throughout the studies within the papers) and how males and females are included in any analyses. Often in clinical studies, sex is used as a covariate, which controls for sex by removing the linear variation due to sex from the analysis and does not inform on the effect of sex. Therefore, in the present study, we examined not only whether a statistical analysis was done in the studies reported in the papers we analysed, but what type of analysis was done (including whether sex was controlled for via a covariate analyses, or explicitly examined as a discovery variable). We were also interested in how many papers reported on studies that used an experimental design that was optimal for discovery of potential sex differences (including reporting sample size, relatively balanced design, using sex consistently throughout the studies within the paper). To analyse whether the authors of the papers considered the possibility of noting sex differences in their data, we focused our analysis on the experimental design and analyses within each paper, with the understanding that not all papers would be designed to address sex differences.

Given the prominent sex differences in neurological and psychiatry disorders, we chose to do a detailed examination of six journals that cover neuroscience and psychiatry. Mandates for inclusion of males and females in biomedical research by CIHR, Horizon Europe and NIH were put in place in 2010, 2014 and 2016, respectively. In order to examine a period from before, and a period after, those mandates were introduced, we examined data from the years 2009 and 2019, as was done previously by Woitowich and colleagues[22]. We hypothesised that there would be an increase in the number of papers that included studies using both sexes from 2009 to 2019 in the neuroscience and psychiatry journals examined, but also that there would not be an increase in papers that had studies that used an experimental design that was optimised to examine sex as a biological variable. We also expected that most papers reporting studies that analysed sex as a factor would do so without using sex as a primary discovery

variable across both disciplines, irrespective of year. Here we show that although the vast majority of papers in this sample reported studies that include both sexes, only 19% included studies using an optimal design for the discovery of possible sex differences and only 5% included sex as a discovery variable in 2019.

## Results

We surveyed 3193 research papers in six journals in 2009 and in 2019 (see Fig. 1). Three neuroscience journals (*Nature Neuroscience, Neuron, Journal of Neuroscience*) and three psychiatry journals (*Molecular Psychiatry, Biological Psychiatry, Neuropsychopharmacology*) were chosen based on high impact factor (ISI) and previous studies[22,26,27]. We determined whether studies within each paper sampled included both sexes and whether the experimental design (balanced, using sex consistently throughout the study) and analyses (sex as a discovery variable) were optimal for discovery of possible sex differences (see details in Methods Section). Proportional variables were used for analyses as the number of papers published differed across journal and year (Table 1).

**Categorisation of papers sampled by reported use of human participants, rodents, or other models**. We categorised the papers in our sample by the subject species or tissue reported (Fig. 2). If a paper used more than one type of subject this was

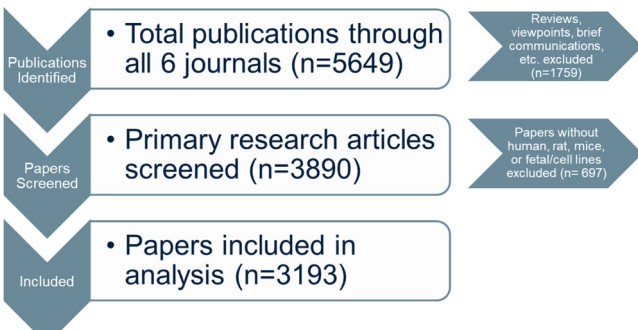

**Fig. 1 Total number of papers sampled in 2009 and 2019 across six journals in neuroscience and psychiatry.** Reviews, viewpoints, brief communications and any other non-primary research articles were excluded. A total of 2456 studies did not match the inclusion criteria and were excluded. Only primary research articles containing human, rat, mice, fetal or cell lines were analysed further ($n = 3193$).

**Table 1 The number of papers examined that were published in 2009 or 2019 in the six journals investigated.**

| Journal | Number of Papers |
| --- | --- |
| *Neuron* 2009 | 159 |
| *Neuron* 2019 | 207 |
| *Nature Neuroscience* 2009 | 118 |
| *Nature Neuroscience* 2019 | 143 |
| *Journal of Neuroscience* 2009 | 1067 |
| *Journal of Neuroscience* 2019 | 588 |
| *Molecular Psychiatry* 2009 | 70 |
| *Molecular Psychiatry* 2019 | 55 |
| *Biological Psychiatry* 2009 | 245 |
| *Biological Psychiatry* 2019 | 136 |
| *Neuropsychopharmacology* 2009 | 209 |
| *Neuropsychopharmacology* 2019 | 196 |
| Total | 3193 |

counted twice (see details in Methods). The majority of papers in the psychiatry journals in this sample reported on studies that used human subjects, but this was closely followed by rodent studies. The majority of papers in neuroscience journals in this sample reported studies that used rodents, which was three times higher than the proportion of papers reporting studies using human subjects. The neuroscience journals in this sample published three times more papers reporting studies using cell lines than the psychiatry journals analysed.

**More papers in the sample reported studies using males and females in 2019 compared with 2009, particularly in neuroscience**. Each paper was examined to determine whether any studies within the paper used both sexes, even if the data were not shown. Across both years of the sample, and both disciplines, the majority of all papers reported studies using both sexes (52.93%), which increased from 37.84% in 2009 to 68.01% in 2019. Overall across both years, less than half (45.28%, $n = 962$) of all the neuroscience papers analysed reported studies using both sexes, while 60.58% ($n = 377$) of all psychiatry papers analysed reported studies using both sexes. Neuroscience papers reporting studies using both sexes significantly increased to 70.39% in 2019 compared with 20.17% in 2009 ($p = 0.003$; Cohen's $d = 9.154$) in the sample studied. Psychiatry papers reporting studies using both sexes increased to 65% in 2019 compared with 55.52 % in 2009 in the sample studied ($p = 0.316$; interaction effect of year by discipline: $F(1, 8) = 8.844$, $p = 0.017$, $n_p^2 = 0.525$ ; Fig. 3a). There were also significant main effects of year ($F(1, 8) = 20.018$, $p = 0.002$, $n_p^2 = 0.714$) and discipline ($F(1, 8) = 5.145$, $p = 0.050$, $n_p^2 = 0.391$).

However, papers in our sample that we identified as including studies using both males and females also included papers that did not disclose sample size, used sex inconsistently within studies, or did not have a balanced design. We therefore also separately quantified papers that not only reported studies that included both sexes, but that also reported sample sizes of males and females, studies that examined sexes using a balanced design, and that consistently used males and females throughout all the studies in the paper. This more stringent quantification of the use of both sexes in the papers identified 16.54% of papers in the sample overall that included studies that used both sexes with an optimal design for the discovery of sex differences (14.15% in 2009 to 18.93% in 2019). Psychiatry papers in the sample (21.40%) were twice as likely to meet these criteria compared to neuroscience papers (11.69%; main effect of discipline, ($F(1, 8) = 11.19$, $p = 0.010$, $n_p^2 = 0.583$). There was no main effect of year ($F(1, 8) = 2.715$, $p = 0.137$, $n_p^2 = 0.253$) or interaction ($F(1, 8) = 0.532$, $p = 0.486$, $n_p^2 = 0.062$; Fig. 3b).

The percentage of papers that did not report sex of subjects used was lower in 2019 (4.24%) compared with 2009 (30.03%, with the greatest change seen in the neuroscience sample (2.81 % of papers omitted reporting of sex in 2019, compared with 52.48% in 2009) ($p < 0.0001$, Cohen's $d = 7.73$). There was no significant difference in percentage of papers in the psychiatry in the sample that did not report sex of subjects in their studies in 2009 (8.13%) versus 2019 (5.66%; $p = 0.632$, Cohen's $d = 0.43$); discipline by year interaction ($F(1, 8) = 45.220$, $p = 0.0001$, $n_p^2 = 0.849$, Fig. 3c). There were also main effects of discipline ($F(1, 8) = 34.975$, $p < 0.0003$, $n_p^2 = 0.813$) and year ($F(1, 8) = 55.200$, $p = 0.0007$, $n_p^2 = 0.873$).

**Most papers in this sample did not use study designs optimal for discovery of sex differences**. What is driving the discrepancy between the fact that majority of papers reported studies using both sexes, but that less than 20% of papers reported studies that used sex optimally for discovery of possible sex differences? There

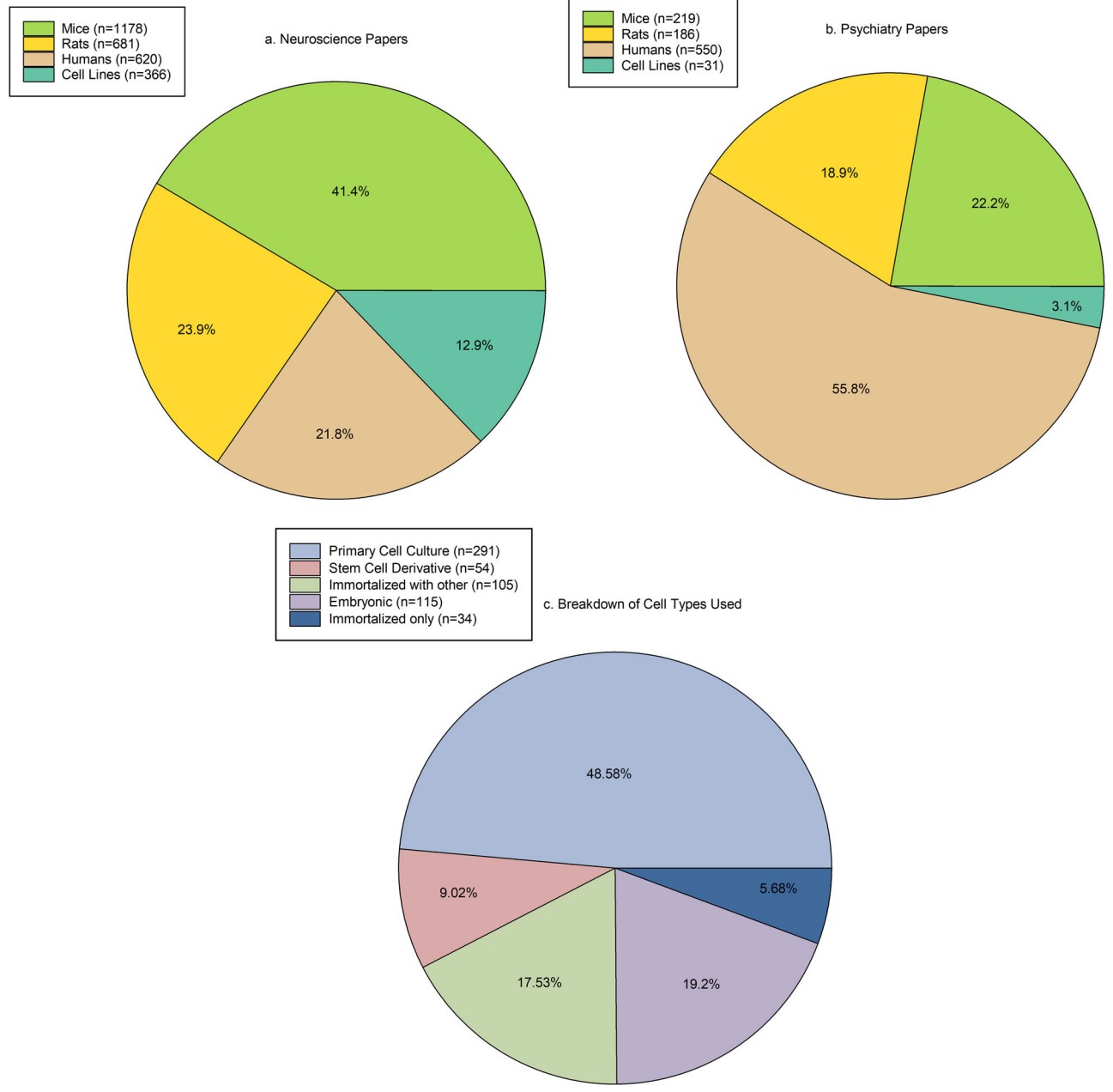

**Fig. 2 Reported species model in each paper sampled across our survey. a** Rodents (mice ($n = 1178$) and rats ($n = 681$)) were the most common species by studies in the papers from the neuroscience journals analysed. **b** Human subjects ($n = 550$) were the most common species used by studies in the papers in the psychiatry journals analysed. Sample sizes ($n$) are the number of papers that reported studies that used the model systems and will total to greater than 3193 as some papers reported studies used two or more model systems. **c** Breakdown of type of cell line used in studies reported in the papers. The largest proportion of papers reported studies that used primary cell lines. Of the papers that reported studies that used cell lines, the majority reported use of primary cell culture ($n = 291$). The other types of cell line reported in the papers were stem cell derivatives ($n = 54$), immortalised with other cell types ($n = 105$), embryonic ($n = 115$), and immortalised only ($n = 34$). Sample sizes are the number of papers and will total greater than 397 as some papers reported studies used two or more cell lines. We relied on the paper to distinguish whether cell lines were conducted in males or females, regardless of the cell line used, and if information was not available on the sex of the cells, they were categorised as sex not reported. Source data are provided as a Source Data file.

were several scenarios we encountered in papers in this sample that reported studies that used males and females but did not use an optimal design. These included: (1) sample sizes were not given (25%); (2) the proportions (>60% in one sex) of the sexes used in the studies within the paper were substantially different (34%); or (3) both sexes were not used consistently throughout the studies within a particular paper (15%, Fig. 3d–f).

Of the papers that used both sexes, just over a third of the papers in our analysis (34.53%) reported studies that did not use a balanced design, with more psychiatry papers in the sample employing this practice (main effect of discipline: $F(1, 8) = 8.189$, $p = 0.021$, $n_p^2 = 0.505$, Fig. 3d). There were no other significant effects (all $p$-values > 0.153).

Just over a quarter of the papers sampled (25.29%) that used both sexes did not identify sample sizes. The percentage of papers

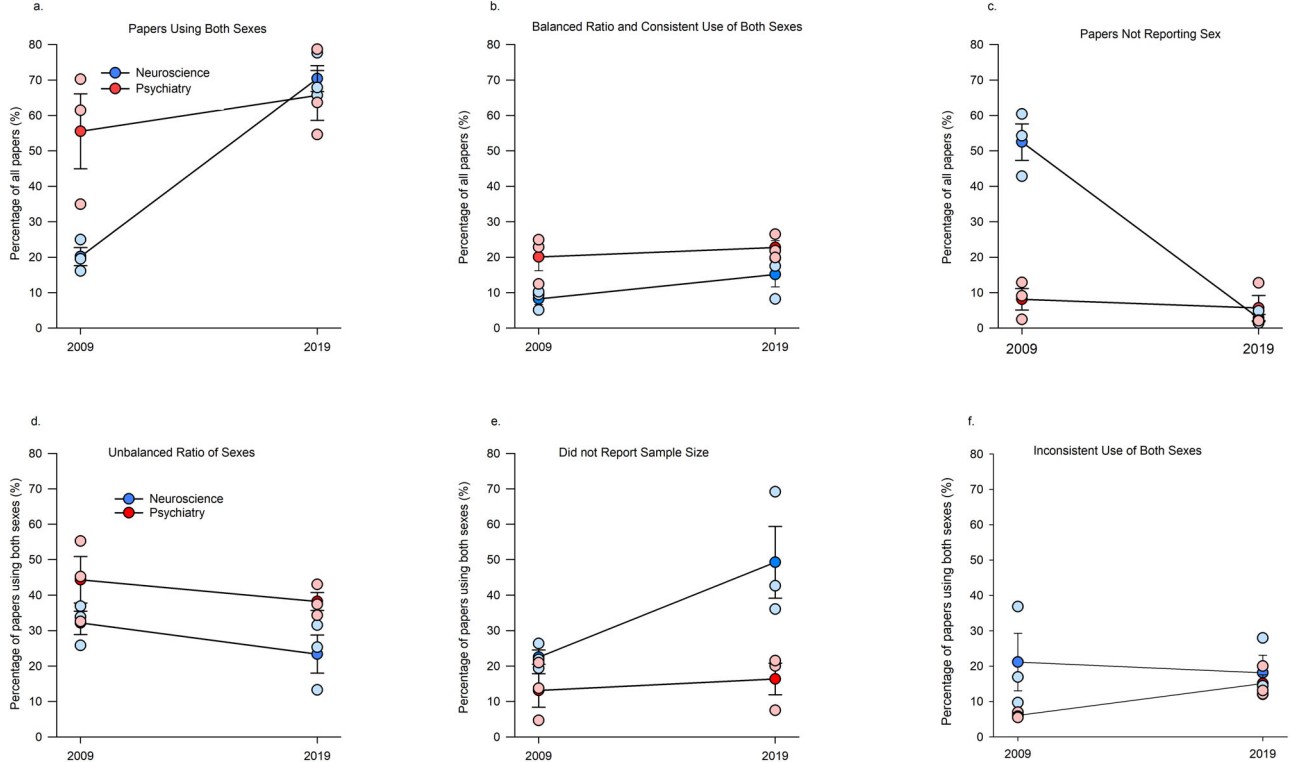

**Fig. 3 The percentage of papers in the sampled in 2009 and 2019 in neuroscience and psychiatry that reported use of both sexes, and the breakdown of how papers were using both sexes.** Plotted are the percentage of proportional papers within each journal and year, n = the number of research papers within each category. **a** Percentage of papers reporting studies using both sexes in any aspect of the paper, regardless of consistency or balanced ratios. The percentage of papers reporting studies including males and females increased significantly for neuroscience (Newman–Keul's post hoc $p = 0.003$, two-tailed) but not psychiatry papers (interaction effect of year by discipline: $F(1,8) = 8.844$, $p = 0.017$, Newman–Keul's post hoc $p = 0.319$, two-tailed). Number of papers: neuroscience 2009 $n = 316$, 2019 $n = 646$; psychiatry 2009 $n = 288$, 2019 $n = 249$. **b** Percentage of papers reporting studies using both sexes consistently throughout the paper with balanced ratios of the sexes. Number of papers: neuroscience 2009 $n = 130$, 2019 $n = 158$; psychiatry 2009 $n = 103$, 2019 $n = 87$. **c** Percentage of papers not reporting sex (sex omission) was decreased in the neuroscience discipline; discipline by year interaction ($F(1,8) = 45.21$, $p = 0.0001$, Newman–Keul's post hoc $p = 0.0002$, two-tailed). Number of papers: neuroscience 2009 $n = 617$, 2019 $n = 25$; psychiatry 2009 $n = 34$, 2019 $n = 14$. Means $+/-$ standard error of the mean. **d** Unbalanced design (i.e., more than 60% of the subjects were one sex) was 34.52% of all papers including both sexes Number of papers: neuroscience 2009 $n = 105$, 2019 $n = 154$; psychiatry 2009 $n = 142$, 2019 $n = 98$. **e** Papers reporting studies using both sexes but not disclosing sample sizes, increased in the neuroscience sample (a priori $p = 0.015$; interaction ($F(1, 8) = 3.73$, $p = 0.089$) but not in the psychiatry sample ($p = 0.717$). Number of papers: neuroscience 2009 $n = 69$, 2019 $n = 304$; psychiatry 2009 $n = 27$, 2019 $n = 38$. **f** Inconsistent use of sex within the studies reported in a paper (i.e., using a balanced ratio in one study within the paper, and an unbalanced ratio or one sex in the other studies within the paper) accounted for 15.11% of papers that we had identified as reporting studies using males and females Number of papers: neuroscience 2009 $n = 55$, 2019 $n = 102$; psychiatry 2009 $n = 17$, 2019 $n = 34$. Means ∓ standard error of the mean. Source data are provided as a Source Data file.

that did not disclose sample size has increased from 17.79% in 2009 to 32.79% in 2019, regardless of discipline. This practice was observed more often among neuroscience (35.87%) compared to psychiatry papers in the sample (14.71%; Fig. 3e; main effects: year ($F(1, 8) = 6.064$, $p = 0.039$, $n_p^2 = 0.431$) discipline: ($F(1, 8) = 12.078$, $p = 0.008$, $n_p^2 = 0.602$). The increase in failure to disclose sample size between 2009 to 2019 is driven by neuroscience papers (49.25% in 2019 compared with 22.49% in 2009; a priori $p = 0.015$, Cohen's $d = 2.16$), whereas the percentage did not significantly differ between the 2009 and 2019 in the psychiatry papers sampled (from 13.09% in 2009 to 16.32% in 2019, $p = 0.717$; interaction ($F(1, 8) = 3.73$, $p = 0.089$, $n_p^2 = 0.318$).

Fifteen percent (15.11%) of the sampled papers that had studies that used both sexes inconsistently used both sexes across the studies within the paper. This percentage did not significantly differ by year or by discipline (Fig. 3f; main effect of year: $F(1, 8) = 0.368$, $p = 0.561$; main effect of discipline: $F(1, 8) = 3.385$, $p = 0.103$, interaction: $F(1, 8) = 1.488$, $p = 0.257$).

**Table 2 The proportional percent of times that male and female data was found in the supplemental section as opposed to in the main body of the paper in our sample.**

| Discipline | Mean ± SEM 2009 | Mean ± SEM 2019 |
|---|---|---|
| Neuroscience | 3.9 ± 2.0 ($n = 3$) | 4.2 ± 4.2 ($n = 14$) |
| Psychiatry | 1.2 ± 1.2 ($n = 6$) | 6.9 ± 2.1 ($n = 16$) |

There were no significant differences by year or discipline (main effect of year: $F(1,8) = 1.342$, $p = 0.280$; main effect of discipline: $F(1,8) = 0.0006$, $p = 0.994$, interaction: $F(1,8) = 1.0893$, $p = 0.327$). Overall four percent of papers sampled referred to data on males and females in the supplemental section. $n =$ number of papers.

Few (4.04 %) papers in the sample referred to the sex effects in the supplemental section and there were no significant differences across year or discipline (main effect of year: $F(1, 8) = 1.342$, $p = 0.280$; main effect of discipline: $F(1, 8) = 0.0006$, $p = 0.994$, interaction: $F(1, 8) = 1.0893$, $p = 0.327$; Table 2).

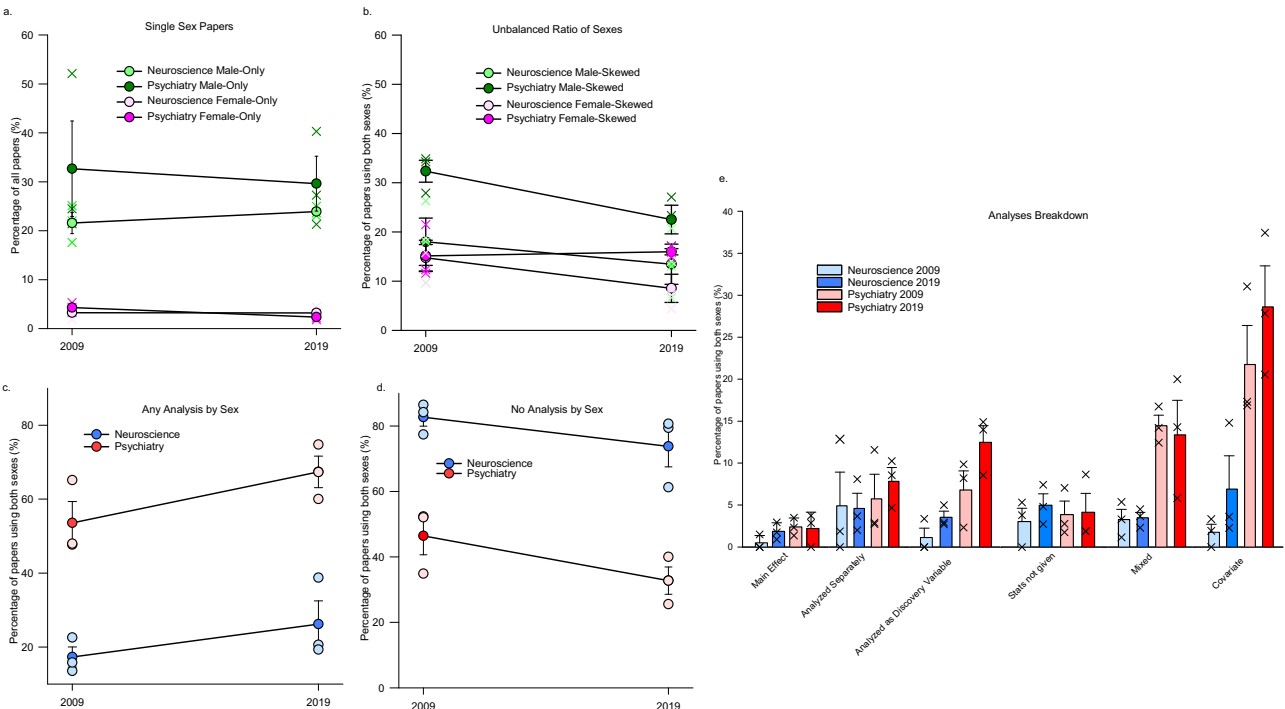

**Fig. 4 Percentage of papers sampled in 2009 and 2019 in neuroscience and psychiatry, which used only a single sex in the paper or which used both sexes, and the breakdown of types of analyses used.** Plotted are the percentage of proportional papers within each journal and year, $n$ = the number of research papers within each category. **a** Percentage of papers reporting studies using only one sex across years and disciplines. Male-only papers (26.96%) were 8.2 times higher than female-only (3.29%) papers (main effect of sex: $F(1,8) = 324.39$, $p < 0.0001$; number of papers: male-only: neuroscience 2009 $n = 322$, 2019 $n = 229$; psychiatry 2009 $n = 184$, 2019 $n = 123$; female-only neuroscience 2009 $n = 55$, 2019 $n = 35$; psychiatry 2009 $n = 23$, 2019 $n = 10$). **b** Of the studies using an unbalanced ratio of sex, there were more studies with greater proportion of males compared to females (number of papers: Male-skew: neuroscience 2009 $n = 56$, 2019 $n = 84$; psychiatry 2009 $n = 97$, 2019 $n = 60$; female-skew neuroscience 2009 $n = 56$, 2019 $n = 70$; psychiatry 2009 $n = 51$, 2019 $n = 39$). As the percentage is proportionally based on the number of publications that year per journal the number of papers will vary differently that the proportional representation. **c** Breakdown of the type of analyses used by papers that used both sexes. Categories of sex analysis include: main effect of sex, sexes analysed separately, sex analysed as a discovery variable, statistics not given (i.e., authors of the paper state some analysis was done but did not provide any statistics) mixed (i.e., any combination of analyses, which may or may not be consistent throughout the study), and sex as a covariate. Number of papers: main effect: neuroscience 2009 $n = 4$, 2019 $n = 12$; psychiatry 2009 $n = 8$, 2019 $n = 5$; sex analysed separately: neuroscience 2009 $n = 9$, 2019 $n = 22$; psychiatry 2009 $n = 12$, 2019 $n = 19$; analysed as discovery: neuroscience 2009 $n = 9$, 2019 $n = 27$; psychiatry 2009 $n = 24$, 2019 $n = 34$; stats not given: neuroscience 2009 $n = 11$, 2019 $n = 32$; psychiatry 2009 $n = 8$, 2019 $n = 7$; mixed: neuroscience 2009 $n = 5$, 2019 $n = 19$; psychiatry 2009 $n = 39$, 2019 $n = 38$; covariate: neuroscience 2009 $n = 6$, 2019 $n = 33$; psychiatry 2009 $n = 54$, 2019 $n = 75$. **d** Majority of papers using both sexes did not analyse by sex, but this decreased slightly over 10 years. Number of papers: neuroscience 2009 $n = 270$, 2019 $n = 498$; psychiatry 2009 $n = 143$, 2019 $n = 76$. **e** Any analysis of sex in studies using both sexes. Psychiatry papers were more likely to perform any type of sex analysis than neuroscience papers. Neuroscience 2009 $n = 46$, 2019 $n = 148$; psychiatry 2009 $n = 145$, 2019 $n = 173$. Means ∓ standard error of the mean. Source data are provided as a Source Data file.

**Nine times more papers reported studies using males only compared to papers that reported studies using females only.** Papers reporting studies using only males as their subjects or participants (which we refer to as "male-only papers" papers) were eight times more common than papers reporting studies using only females as their subjects (which we refer to as "female-only papers"), regardless of year (main effect of sex: $F(1, 8) = 324.39$, $p < 0.000001$, $n_p^2 = 0.976$; Fig. 4a). The percentage of papers in our sample that reported studies that only included one sex did not significantly differ between 2009 and 2019 (27% in males, 3% in females; $p = 0.359$, Cohen's $d = 0.0359$) or across disciplines ($p = 0.340$, Cohen's $d = 0.932$).

Of the papers that reported studies that used males and females in an unbalanced design (which we refer to as "sex-skewed papers"), almost twice as many reported studies using more males (21.58%) than those that reported studies using more females (13.61%; main effect of sex skew: $F(1, 8) = 20.230$, $p = 0.002$, $n_p^2 = 0.717$) and there were almost double the percentage of sex-skewed papers in the psychiatry sample compared with the

neuroscience sample (main effect of discipline $F(1, 8) = 9.017$, $p = 0.017$, $n_p^2 = 0.531$). There were no other significant effects ($p$-values >0.121); Fig. 4b).

We did a thematic analysis on the reasons that were given within a paper as to why single-sex studies were used. This analysis revealed 51 documented reasons, most of which referenced the need to reduce variability or confounds (50.98%, Table 3).

**The majority of papers in the sample did not include analyses by sex.** Of the papers in the sample that indicated that studies used both sexes, 40.34% reported an analysis of the data by sex. The percentage of papers that reported an analysis by sex increased to 46.36% in 2019 compared with 34.32% in 2009, irrespective of discipline (main effect of year: $F(1, 8) = 5.236$, $p = 0.050$, $n_p^2 = 0.395$). However, psychiatry papers sampled were three times more likely (60.46%) to report an analysis by sex compared to neuroscience papers sampled (21.75%; main effect of

**Table 3 Thematic rationales given in papers sampled that had studies that included one sex in study design.**

| Reason | Proportion |
|---|---|
| To reduce confounds/variability/hormones | 50.98% ($n = 26$) |
| Behaviour (i.e., aggression/fighting) | 15.69 % ($n = 8$) |
| To avoid sex differences | 11.76% ($n = 6$) |
| Disease prevalence | 11.76% ($n = 6$) |
| Lack of previously observed sex difference | 5.88% ($n = 3$) |
| Insufficient offspring | 3.92% ($n = 2$) |

Half (50.98%) of the sampled papers using studies that used a single sex claimed the reason was to reduce confounds or variability mainly due to fluctuating hormones. $n$ = number of papers.

discipline: $F(1, 8) = 61.014$, $p = 0.00005$, $n_p^2 = 0.884$). There was no significant interaction ($p = 0.637$; Fig. 4c).

Overall, whereas the majority of papers in the sample reported studies that used both sexes, the majority of these papers did not include studies that analysed by sex (58.89%). Neuroscience papers reporting studies using both sexes were almost twice as likely to not include analyses by sex (78.24%), compared with the psychiatry papers sampled (39.53%; $F(1, 8) = 61.014$, $p = 0.00005$, $n_p^2 = 0.884$). The percentage of papers reporting studies not including analyses by sex significantly decreased in 2019 to 53.22% compared with 64.56% in 2009 ($F(1, 8) = 5.240$, $p = 0.050$, $n_p^2 = 0.396$; Fig. 4d).

**Six percent of papers reporting studies using both sexes used an optimal analysis of possible sex differences.** We further broke down how the papers that included studies that analysed by sex into 6 categories: complete analysis by sex (analysed as a discovery variable), statistics not given, covariate, main effect, analysed separately, and mixed analysis (see details in Methods). Of the papers that reported studies that used both sexes across both years and disciplines, 6.00% used sex as a discovery variable in our sample. Of the papers that reported studies that used both sexes, 14.36% used sex as a covariate in our sample (Fig. 4e).

Psychiatry papers sampled were 5 times more likely to include studies that used sex as a covariate ($p = 0.0001$, Cohen's $d = 2.998$) or a mixed analyses ($p = 0.003$, Cohen's $d = 2.989$) compared to neuroscience papers sampled, regardless of the year (analyses type by discipline: $F(5,40) = 10.231$, $p = 0.00002$, $n_p^2 = 0.56$)). Covariate analyses were more often used than any other analysis ($p$'s $< 0.001$; main effect of Analysis Type: $F(5,40) = 13.140$, $p < 0.001$, $n_p^2 = 0.622$). There was also a main effect of discipline ($F(1, 8) = 60.274$, $p = 0.00005$, $n_p^2 = 0.883$) and a main effect of year with 2019 being higher than 2009 ($F(1, 8) = 5.170$, $p = 0.050$, $n_p^2 = 0.393$), but no other significant effects ($p$'s $> 0.430$, $n_p^2 < 0.111$).

**Papers from groups based in North America showed a slight increase in inclusion of analyses by sex from 2009 to 2019.** We next examined institutional affiliation of all authors of each paper to determine country of origin of the papers sampled (see details in Methods). We did an analysis using the proportional data based on the number of papers that used studies with both sexes by the country of author institutional affiliation (using each country as its own baseline) across both years for the EU, UK, Canada, USA, Asia and when there was a combination of countries within the authors affiliations (Fig. 5a–f). There were low percentages across all countries with no significant difference across countries by year ($F(5, 35) = 0.903$, $p = 0.490$, $n_p^2 = 0.114$) or by discipline ($p$-values $> 0.152$, $n_p^2 = <0.269$; Fig. 5g–j). There

were no other significant effects ($p$-values $> 0.315$; number of papers Supplementary Table 1). When comparing between 2009 to 2019 (compare Fig. 5g to 5h), there was a non-significant increase in the relative percentages in papers that used an optimal analysis for discovery of possible sex difference from groups based in the USA, Canada and a combination of countries, but there was a corresponding decrease in the relative percentages in papers from research groups based in the EU, Asia, and the UK across the same time period.

**Female-inferred authors on the paper were associated with a greater proportional increase in analysis by sex compared to male-inferred authors.** As previous studies have noted that the analysis and inclusion of sex as a variable in papers may be related to inferred author sex[28–30], we examined whether inferred-sex of the first or last author was associated with the percentage of papers that used sex as a discovery variable, an optimal design for discovery of possible sex differences, or single-sex papers (see Methods for details). As these estimates are based on names and not on self-reported gender identity we use the term "inferred-sex" when referring to the authors. A greater proportion of papers that listed female-inferred first author names compared to male-inferred first author names considered sex as a discovery variable compared to papers that used an optimal design for the discovery of possible sex differences within their studies ($\chi^2 = 5.99$, $p = 0.014$; Supplementary Fig. 2a–d). Moreover, a greater proportion of female-inferred last author names were associated with more papers using female-only compared to male-only subjects/participants compared to male-inferred last names in our sample ($\chi^2 = 2.659$, $p = 0.050$, one-tailed) (Supplementary Fig. 1c, d).

## Discussion

Our survey of 3193 papers across six journals in neuroscience and psychiatry revealed some insights into the inclusion, use, and analyses of both sexes in research in the sample years, 2009 and 2019 (Fig. 6). Most papers in our sample had studies that used both males and females in 2019, a 30% increase from 2009, irrespective of discipline. On the face of it, this is a positive indication of greater knowledge and awareness on the importance of sex and gender as variables in research. However, we found the majority of papers we sampled in 2019, did not use what we consider an optimal design or analysis for the discovery of possible sex differences. This is concerning, as scientific discovery will lose out on valuable information if researchers are neglecting to embrace the power of studying potential sex differences. Specifically, we determined that out of the total number of papers sampled that reported studies using both males and females, 16.5% reported using an optimal design for discovery of sex differences. Most of the sampled papers that used both sexes (75%) either did not specify sample size, used unequal proportions of the sexes, or used the sexes inconsistently within the studies in the paper and furthermore, 58% of these papers did not include an analysis by sex. Only 6% of the total number of papers reporting studies using both sexes included sex as a discovery variable, and this value was consistent across years and disciplines.

Furthermore, the percentage of papers reporting studies using optimal designs or analyses for discovery of sex differences has not meaningfully shifted between 2009 and 2019 across either discipline, despite the number of recent initiatives such as SABV, SGBA and SAGER[14,16,31]. It is possible that these percentages will increase with time as these mandates are relatively recent. These findings should serve as a reminder to researchers, funders and publishers, that if we are to harness the wealth of knowledge from

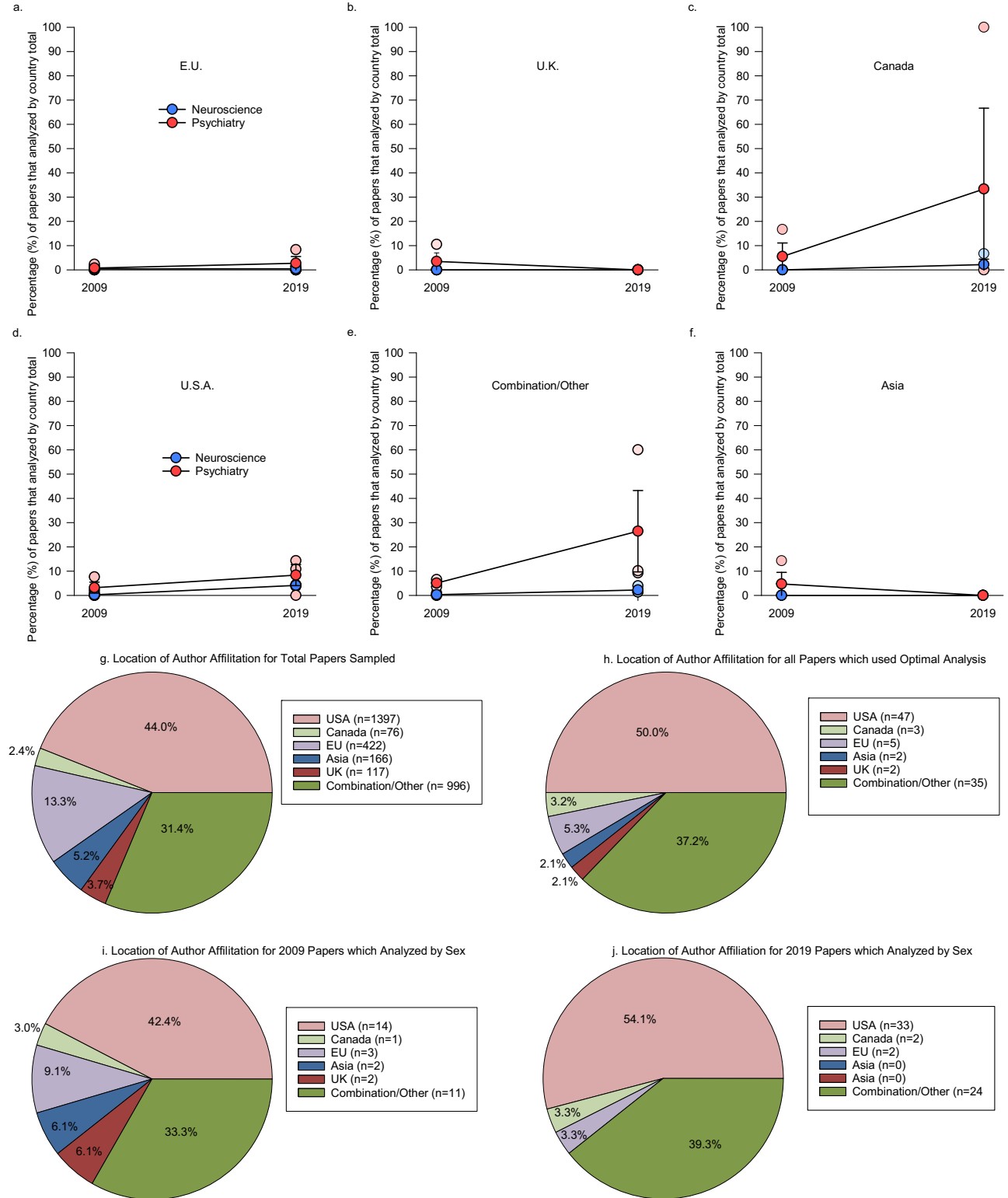

studying the sexes, more needs to be done to improve the appropriate application of sex in reporting and analyses for discovery.

As noted, there has been an increase in the reporting of both sexes in both psychiatry and neuroscience papers in our sample to almost 70% in 2019 from 20% in 2009. The neuroscience sample showed a 50% increase in reporting the use of both sexes over the years whereas the increase between 2019 and 2009 was only 10% in the psychiatry sample. This difference between the disciplines

is likely driven by the majority of papers in the psychiatry sample using humans as participants, which may be a direct result of an earlier (2001) NIH mandate to include males and females in clinical research. The majority of neuroscience and psychiatry papers in our sample reported studies that used both sexes in 2019, which is encouraging. Our finding of a 50% increase in the 2019 neuroscience sample compared to 2009 sample is higher than previous work[22,23], indicating an upward trend over the years sampled. For example, past research sampling neuroscience

**Fig. 5 Institutional author affiliation of the authors of each paper sampled in neuroscience and psychiatry in 2009 and 2019.** If a different country was noted among the author affiliations within a paper, this was considered as a combination of countries. Plotted are the percentage of proportional papers within each journal and year, n = the number of research papers within each category. **a–f** Country or combination of countries of author affiliations and the respective percentage of papers reporting studies that analysed using sex as a discovery variable across years compared to the country total. Papers from research groups based in the USA, Canada, EU and a combination of countries had an increased percentage of studies that analysed by sex as a discovery variable but none of these were significant. Means ∓ standard error of the mean **a** E.U. is the European Union (number of papers: neuroscience 2009 $n = 2$, 2019 $n = 1$; psychiatry 2009 $n = 1$, 2019 $n = 1$. **b** U.K. is the United Kingdom (number of papers: neuroscience 2009 $n = 0$, 2019 $n = 0$; psychiatry 2009 $n = 2$, 2019 $n = 0$. **c** Canada (number of papers: neuroscience 2009 $n = 0$, 2019 $n = 1$; psychiatry 2009 $n = 1$, 2019 $n = 1$. **d** U.S.A. is the United States of America (number of papers: neuroscience 2009 $n = 4$, 2019 $n = 15$; psychiatry 2009 $n = 10$, 2019 $n = 18$. **e** Combination of countries: (number of papers: neuroscience 2009 $n = 3$, 2019 $n = 10$; psychiatry 2009 $n = 8$, 2019 $n = 14$. **f** Asia: (number of papers: neuroscience 2009 $n = 0$, 2019 $n = 0$; psychiatry 2009 $n = 2$, 2019 $n = 0$. **g** Country/region of author affiliation of all papers sampled and **h** breakdown of papers sampled that had studies using optimal analysis for discovery of sex differences by country or region of author affiliation. **i–j** Breakdown of papers reporting studies, which analysed by sex by country or region of author affiliation in 2009 (**i**) and 2019 (**j**) Means ∓ standard error of the mean. Source data are provided as a Source Data file.

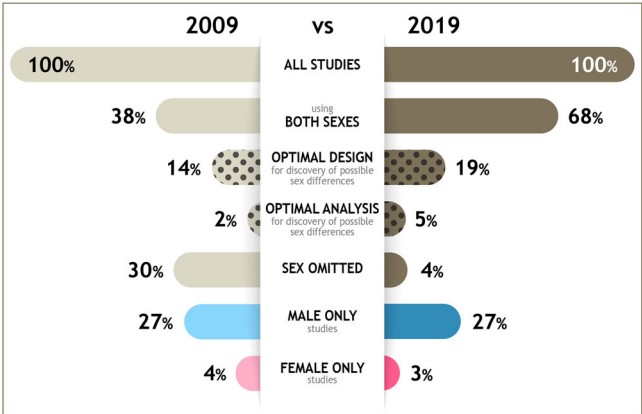

**Fig. 6 An infographic depicting the change in percentages of total papers sampled reporting studies in 2009 and 2019 that used both sexes, a single sex, omitted sex, papers reporting studies that used an optimal design or analyses for the discovery of possible sex differences irrespective of discipline.** Optimal design refers to relatively based sample size and use of males and females consistently across the experiments whereas optimal analyses refers to the use of sex as a discovery variable. Although the percentage of studies in the sample of neuroscience and psychiatry papers analysed has increased the use of optimal design and analyses has not changed as much and remain at low levels. There are nine times the percentage of male-only compared to female-only studies.

papers reporting the use of both sexes has found an almost 20% increase between 2010–2014[25] and a 34% increase across the same time points assessed here[22]. In addition, the 68% of papers reporting studies that included males and females in 2019 in our study is higher than the 52% of neuroscience papers reporting the use of both sexes in 2017[23], again likely reflecting an upward trend across years. The large progress made in neuroscience across the 10 years was also noted by Woitowich[22] who reported an increase to 63% in 2019 using a sampling of 20 articles from 4 journals, two of which overlapped with ours (*Journal of Neuroscience* and *Nature Neuroscience*). In the present work we sampled from 3 journals in neuroscience, similar to Meitzen and colleagues[23,27] who sampled neuroscience papers in 6 journals, 3 of which overlapped with the journals we chose (*Nature Neuroscience*, *Neuron*, *Journal of Neuroscience*). Thus, collectively, multiple studies, using different journals and methods of sampling, consistently indicate that there is an increasing trend in papers that include males and females in their work.

Although the use of both males and females in research has been steadily increasing to include a majority of studies, research highlighting or mentioning sex differences is scarce. Why might this be? We examined whether papers were reporting studies that

used optimal designs for discovery of possible sex differences. When we accounted for papers in our sample that did not disclose sample size of the sexes, used an unbalanced design or did not use both sexes throughout all the studies within the paper, we found that only 16% of papers used a design that was optimal for discovery of sex differences. Some researchers may argue that investigating both males and females is only important in the first study of the paper and thus the use of both sexes in further experiments, beyond the initial study is not required. However, there are numerous examples where a trait may not show sex differences but the neural mechanisms underlying that trait do show significant differences between males and females[32–35]. Thus, using males and females in one experiment does not preclude the fact that sex differences may be seen in further related experiments that uncover mechanism. The use of what we consider to be the most advantageous design for discovery of sex differences was employed in just under 20% of papers sampled in 2019. Thus, although it appears on the face of it that most papers sampled report studies using males and females, the majority of these studies do not incorporate sex in a design that we consider optimal for discovery of possible sex differences.

Our findings also demonstrated that 25% of the papers sampled that report studies using both males and females do not report sample size, consistent with the findings from Woitowich and colleagues[22]. Perhaps more concerning is that in the neuroscience papers sampled, this trend increased between 2009 to 2019 with almost 50% not reporting the sample size of males and females used in 2019. This trend is troubling as readers are unable to judge how effectively males and females were used in the study.

As others have reported[21–23,26,27], most publications do not report studies that analyse by sex. In our analysis, only 6% of the papers in our total sample that reported studies that used males and females also used sex as a discovery variable, which did not increase in the 2019 compared to the 2009 sample. This translates into only 4% of all the publications examined in our sample from both years and both disciplines that used sex as a discovery variable, matching a previous estimate[21]. The most common statistical method for analysing sex that was used by papers in our sample was controlling for sex using a covariate. A covariate removes the linear association of the factor of sex against the dependent variable, removing any linear variation due to sex. In our view, this is in opposition to the intention of SABV or SGBA, as we believe that the goal of these mandates is not to remove the variation due to sex but to determine whether or not sex is a variable that could be causing differences in outcomes. The use of sex as a covariate can result in the reduction of power and the loss of important information when a sex difference is present[36]. Mersha and colleagues[36] show that 26 more single-nucleotide polymorphisms (SNPs) were identified in a sex-stratified analysis compared to when sex was used as a covariate. Put another way,

when sex was used as a discovery variable 47 SNPs were identified that were associated with asthma but if sex was used as a covariate only 21 SNPs were identified[36]. They also found that effect sizes were larger when a sex-stratified analysis was used, contrary to popular opinion that power would be negatively affected with the addition of sex as a discovery variable. Some argue that design and sample sizes are not sufficiently powered to consider sex-stratified analyses, but if the sex effects are large, or in opposing directions, the resulting power with the inclusion of sex as a discovery variable, will improve as others have demonstrated[25,26,36,37]. Taken together, our survey of the literature suggests that researchers are underestimating the power of using sex as a discovery variable in their research.

Similar to other reports in neuroscience and other biological disciplines[23,24,27,37], we found that papers we identified as "female-only" (i.e., reporting studies that used females only) were a small percentage of the sample (3%)[38,39]. Our findings are comparable with others showing that 5% of neuroscience papers sampled were female-only in 2009[37] and in 2017[23]. While the consideration of sex and gender in studies is important, in our view, single-sex studies are still needed. In particular, given the dearth of information on women's health, disparities in diagnosis[7], and continued underrepresentation of women in clinical trials[13], one could argue that female-only studies are needed more so than male-only studies—or at least that single-sex studies should be conducted and published in equivalent proportions. Indeed, mandates such as SABV and SGBA were instigated in part because of the lack of knowledge of how females differed in their response to treatments and disease[17]. There are female-specific experiences that affect female health, such as menstruation, hormonal contraceptives, pregnancy and meno-pause that need to be studied[40–43]. Unfortunately, as highlighted by the current study, the percentage of papers that use only females remains consistently low and has not increased in 2019 compared with 2009 in our sample. Funders and researchers should work to correct this imbalance.

Our current study indicated that the rationale for excluding females used most often was to "reduce variability". To exclude females based on greater variation than males is not valid, as two studies have found that the within-sex variability, on a variety of measures, is not statistically different between males and females in rats and mice[44,45]. Moreover, our findings and others[37], reveal that it is a common belief that females will have more variability due to their hormones, however it is important to note that both males and females have diurnal fluctuations in cortisol or corti-costerone in humans and rodents, respectively[46,47]. Furthermore, human males have diurnal fluctuations in testosterone levels that vary significantly with age[46] Researchers should be encouraged to consider that many hormones vary with diet, age, housing con-ditions, and experience across both sexes[48–50]. Thus, variability within females should not be considered a limiting factor to the use of females in research[51].

There have been calls in the literature for editors and reviewers of manuscripts to ensure that published reports use both males and females and report on outcomes[51]. SAGER guidelines were developed by the European Association of Science Editors to improve sex and gender in research reporting in 2016[31], and indeed, some journals have adopted SAGER guidelines including over 500 Elsevier journals[52]. Among the guidelines, it is recom-mended that authors include the sex or gender in the title and abstract, background information on sex and gender effects on the variables of interest in the paper and in the results to dis-aggregate and analyse the data by sex or gender. However, the percentage of journals that have adopted SAGER is low with one study finding under 10% of journals in psychology had adopted the guidelines[53] and in those journals the guidelines were only

adopted for the title, abstract and methods but not on reporting of analyses or data by sex or gender[53]. However, as can be seen from the present data, the publishing of this information, parti-cularly with respect to the analyses of sex as a discovery variable is limited, and a more concerted effort needs to be adopted.

We note several limitations to this work. We only examined three journals for each of the two disciplines, however we did a comprehensive search of eligible research papers within each journal, culminating in over 3000 papers reviewed. Previous studies have either surveyed 841 articles across the same 2 years[22] or examined 6000 articles across 4 years[27]. We, like others[27], selected journals based on high ranking by ISI, with some overlap in journals chosen. However, our comprehensive search of these six journals gave values that were not appreciably different from those that used fewer papers within more journals, or those that carried out analysis of more publication years. This suggests that the different survey methods used across these bodies of work yield similar results.

A final consideration is that for biomedical research at NIH, the SABV consideration was instituted in 2016, which may not have given enough time to fully realise the potential in the 2019 sample examined. However, the fact that, in the neu-roscience journals sampled here, there was a 70% increase in 2019 in the percentage of papers with studies using both males and females, suggests there is greater inclusion of studies using males and females. However, this increased use of both sexes in studies is unfortunately not yet resulting in using sex as a discovery variable in analyses.

Given that there is excellent uptake in the use of both males and females in research, what is driving the lack of optimal design and analyses for discovery of sex differences? It seems possible that researchers themselves are not aware that they are not using best practices, perhaps due to the lack of consensus on how to use sex in analyses and the required sample size in the literature[17]. In one manuscript on this topic it was reported that three-quarters of researchers say they report the sex in their papers[54], and 50% of these researchers said they analysed their findings by sex[54]. Our results from the literature survey show that although 40% of papers we sampled included analysis by sex in some fashion, only 6% used sex as a discovery variable. Taken together, the findings here, along with prior data[5], suggest that researchers may be considering analyses that are suboptimal or not reporting ana-lyses even when they have done them. Thus, it is possible that researchers believe that the addition of both sexes without thor-ough analyses is enough to satisfy the initiatives.

Researchers themselves may need more training in sex and gender analyses. Qualitative analyses from structured interviews with US-based researchers[55] found that while researchers indi-cated they had a good knowledge of SABV they incorrectly used the terms sex and gender when discussing their views, indicating a lack of knowledge. The misuse of the terms sex and gender has also been noted in grant submissions as well as in the biological literature[56–58]. Gender is a psychosocial construct that includes gender identity and societal expectations for roles and behaviour based on gender identity. Gendered effects can be realised when considering a number of intersectional variables, such as race, ethnicity or age, along with sex and gender identity[59]. Thus, perhaps more training for researchers may be needed to ensure fruitful addition of sex and gender in research.

One could argue that the mandates do not go far enough and are limited to a few agencies in the EU, Canada, and the US. There are also no repercussions when authors do not publish or analyse by sex. Indeed, NIH funding did not significantly affect the percentage of papers that reported studies that analysed by sex with a net increase of just 3% (to 9%) overall[23]. Our data indicate that there is a non-significant increase in the sampled

papers that reported studies that used sex as a discovery variable from research groups based in the US, Canada, and the EU in the 2019 sample studied compared with 2009, pointing to an overall potential benefit of the current mandates that exist in those countries. However, it is important to underscore that the percentage of these papers was low even when the research groups were based in those countries, and that there are no reporting requirements from these funding agencies.

What can funders do to promote more work on sex differences? One solution is to have funding dedicated specifically for SABV and SGBA proposals and not as a supplement to regular funding. Evidence suggests that this approach has been successful in cardiovascular research. For example, the American Heart Association (US) has dedicated funding for sex differences, and as a result sex and gender-based research and analyses in cardiovascular disease has flourished[60]. Our view is that funders should make these funds a significant portion of the budget to provide enough incentive to encourage researchers to think deeply about incorporation of sex in research. Dedicated funding would not only generate proposals and knowledge dedicated to the analyses of sex differences, but they would also have the by-product of creating the next generation of researchers that integrate sex into their research. One can also look at how significant funding to amyotrophic lateral sclerosis (ALS) and AIDS has advanced research in these areas. In 2014, the ice bucket challenge raised greater than $115M in the US and this attention leveraged dedicated funding from other sources tripling ALS research budgets in 5 years[61]. This bolus of funding doubled the number of ALS publications, led to a 50% increase in investigators interested in ALS, and has dramatically accelerated the number of clinical trials in ALS[62]. Scientific evidence takes time to build, but fruits of discovery with the increased funding are paying off with promising new treatments[63]. It's hard not to get excited about the possibilities if this type of funding is extended to fill the sex disparities in health research. AIDS research is another success story with dramatic advancements in AIDS research that came with dedicated funding. Worldwide HIV/AIDS research funding more than doubled from 2000 to 2019 to >1B[64]. With these dedicated funds have come advancements in therapeutics such that individuals with HIV can live relatively full lives[65]. To make significant progress, funders need to have dedicated funding for SABV, which would have a cascading effect to get more researchers interested in SABV, ensure consideration of sex and gender as discovery variables, increase the number of discoveries and train the next generation of SABV researchers.

What can publishers do to promote publications using sex-based analyses? When journals adopt SAGER guidelines, it is up to the authors, reviewers and editors to ensure the guidelines are met. In over a third of submissions to a neuroendocrinology journal, authors and reviewers failed to notice that neither sex nor gender had been disclosed[66]. This suggests, not surprisingly, that not every reviewer is prompted to think about the consideration of sex in experimental design and analyses upon reviewing a paper. Training modules from funders or scholarly organisations with an SABV focus may help, but working on a similar premise as above, enticing researchers to explore the influence of sex and gender in their data may be a more fruitful approach. If journals, especially those with higher visibility, adopt calls for papers using sex and gender-based analyses this may serve as a catalyst to ensure more researchers consider possible sex differences and further promote the notion that this research is important to publish.

Lastly, others have found that the presence of inferred-female first or last authors (inferred from names) was associated with the use and analyses of both sexes in research[28,29] consistent with our own data. Recently, there have been concerted efforts to promote

diversity in science[67] and these findings suggest that increasing sex and gender diversity among authors of scientific research is another fruitful path to improve the percentage of papers reporting use of sex as a discovery factor in analyses.

We hope these data are a call to the research community to not only include males and females in their research but to ensure appropriate methods of integration and analyses are used as well. If researchers are merely including a few animals of the opposite sex in one of many experiments this will not allow for discovery of the impact of sex as a biological variable. Nor will the non-robust adoption of sex in experiments harness the additional power that the analyses of sex can afford[36]. Research shows us that the use of sex as a discovery variable can lead to fruitful knowledge, and can enable conclusions that the different mechanisms between males and females require distinct treatment[25]. Indeed, inclusion of sex in analyses and design will improve not only the health of females but of males as well[68]. We lose collectively, not just in knowledge gained, but also in our search of more effective treatments when sex is not considered in the design and analyses of our studies. We call on funders, reviewers and researchers to recognise that sex and gender matter across all disciplines. The community needs to be aware that there are many types of sex differences[19,69] and that some sex differences are revealed due to perturbations in environment, genotype, or disease[19,70,71] so it is important to continually examine and analyse both sexes throughout the studies. It is imperative that more attention is paid to the appropriate design and analyses of sex and gender in the literature. We need to study how mandates can improve adherence in both study design and dissemination. To ensure precision medicine, we need the community of funders, researchers and publishers to embrace the addition of SABV, SGBA and SAGER to improve the health of women, men and gender-diverse individuals.

## Methods
We examined research papers within three journals in neuroscience and psychiatry across two years. We chose journals based on the top ISI Clarivate rankings that published primary research papers and subject-specific journals within the neuroscience and psychiatry domain, as well as the top society journals (*Society for Neuroscience, American College of Neuropsychopharmacology*) and journals that were chosen in prior research[22,26,27]. Three neuroscience journals (*Nature Neuroscience, Neuron, Journal of Neuroscience*) and three psychiatry journals (*Molecular Psychiatry, Biological Psychiatry, Neuropsychopharmacology*) were chosen. We assessed papers published in the year 2009 and in 2019 to assess whether there has been an increase in the inclusion of sexes, improvements to experimental design and analyses to examine potential differences between the sexes in the studies reported in those papers.

**Studies included**. All primary research papers from 2009 and 2019 were analysed if the papers used rats, mice, human subjects, fetal cells or cell lines were included. Cell lines included immortalised cell lines, primary cell culture, and stem cell derivatives. As sex of cells matters in a variety of outcomes[72] papers that used these tissues were included. Non-primary research articles such as reviews and viewpoints were excluded as well as brief communications due to their brevity and following previous literature[22,23]. This resulted in a review of a total of 3193 papers (Fig. 1). Assessments were done by two trained curators who had >99% interrater reliability (RKR, TFLS). When the categorisation of analyses within the paper (see below) was questioned, these were confirmed by AYA, a biostatistician—who was consulted on 0.5% of the papers reviewed or 16 times in total.

**Categorisation of inclusion of males and females and sex-based analyses**.
Papers reporting studies that matched the inclusion criteria were first examined to determine whether they included males and females, males only, females only, did not report sex, or were inconsistent throughout (i.e., if the studies used males in one experiment, and both sexes in another experiment). If a paper reported a study that used both sexes, we determined whether there was balanced design (an equal ratio of male to female subjects). An unbalanced design was defined as one sex accounting for more than 60% of the total sample size. When a paper reported studies that employed a relatively equal sample size of both males and females (the sample size of one sex was not more than 60% greater than the other) and used them consistently throughout the studies within a paper we refer to this as an optimal design for discovery of sex differences. Our reasoning behind this is that

because unequal sample sizes affect power (the chance that the study will detect a sex difference if a sex difference exists or rejecting the null hypothesis when it is false) and if unequal sample sizes are paired with heterogeneity of variance this will affect the robustness of parametric tests[73]. This underscores that relatively equal sample sizes are necessary for an optimal design for discovery of possible sex differences. Modelling of sample sizes needed for discovery of sex differences suggest that when an interaction is present (interaction is when factor A has a different effect dependent on sex), high power can exist depending on the effect size of the interaction. For example, using a factorial ANOVA, high power (i.e., $\beta > 0.8$) is obtained with relatively small sample sizes ($n = 5$ per group) when the interaction shows either a reverse effect between sexes or no effect in one sex versus the other[25,28,37]. Larger samples sizes are needed when an interaction exists due to half of the effect in one sex over the other sex ($\beta > 0.8$, $n = 25$ per group)[25]. Indeed the use of sex as a discovery variable can lead to increased statistical power, particularly when there are interaction effects indicating the sexes show opposing effects of a treatment or intervention on the variable of interest[25,37]. Thus, it is important that researchers not just consider that sex differences may result in an overall (main) effect but that they may result in interaction effects (when a treatment has different effects in one sex versus another).

Papers reporting on studies that included both sexes were then examined to determine whether they included any form of analysis using sex as a factor. Papers reporting on studies that did any type of sex analysis were then broken down into six categories: main effect of sex only, complete analysis by sex, sex as a covariate, analysed sexes separately, statistics not given, and "mixed analysis". Papers reporting studies that only tested for a main effect of sex (examining differences between males and females on the dependent variable of interest) without regard to whether there were any interactions with other independent variables or any other further analyses were classified as "main effect". An interaction effect examines the effect of sex along with other independent variables (e.g., treatment, genotype, disease). A significant interaction will indicate that the effect on the dependent variable (e.g., neurogenesis) varied across two independent variables, such as neurogenesis levels would differ by drug treatment based on the sex of the subject. Papers reporting studies that analysed the main effects and interaction effects of sex were classified as "complete analysis by sex". Papers reporting studies that used sex a covariate effectively removes the linear association of the variable sex from the dependent variables of interest. A covariate is a way of eliminating the variability due to sex, not analysing for sex, and in doing so covariates are often referred to as "nuisance" or "confound" variables. Some papers stated that there was or was not an effect of sex in their studies but provided no statistical evidence to back up the statement and these papers were classified as "statistics not given". A "mixed analysis" category was also included which consisted of papers that were inconsistent in their analyses throughout the studies reported in the paper (i.e., analysed sex in one experiment but did not analyse by sex in subsequent experiments). Any papers reporting studies that used both sexes but did not mention any effects or analyses by sex and therefore did not fit into any of these "analysed" categories were classified as "not analysed". When sex information and analyses were only reported in the supplementary section of the paper, these papers were put into a "supplementary only" category. When a paper was classified as "analysed" by using sex as a discovery variable, this meant that in the studies reported in that paper, sex was used as a predictor/between-subject variable in the analysis and analysed for main and interaction effects. We refer to this as an optimal analysis for possible discovery of sex differences.

We refer to these designs and analyses as optimal for discovery of possible sex differences, as it would be impossible to detect any sex differences if the data were not analysed by sex and if the sexes were not used consistently or the sample size employed was not advantageous to the discovery of possible sex differences. We do not mean to imply that the studies were not optimal in the design for the particular experiment but that the design or analyses were not optimal for the discovery of any possible sex differences.

The country or region of origin of research groups of each paper was also examined by noting the region or country of the institute affiliation of all the authors. We included six categories for region: USA, Canada, Europe (EU), the United Kingdom (UK), Asia (all countries in the continent of Asia), and combination/other. Combination/other refers to studies done by researchers based in multiple countries or other meaning another region than those listed by institute affiliation.

We also examined the inferred-sex of the first and last author of each paper that used an optimal design, an optimal analysis for discovery of possible sex differences, male-only or female-only papers. We used the website genderize.io, a database, which determines the inferred-sex or gender of a first name and provides a certainty factor associated with the name and has a low error rate for misclassification[74]. When this was not possible (~50% of the time), we inferred author gender by searching for the author online and looking for descriptions of their pronouns using LinkedIn or institutional websites. Given that these websites may mislabel the sex and gender of authors in our sample, coupled with the fact that it is not possible to determine sex or gender identity from names alone, we are not certain whether these names reflect a person's sex or gender. Thus, we use the term inferred-sex to describe authors. When it was not possible to determine inferred-sex, we categorised these as unsure and these authors were included in the analyses.

**Statistical analyses.** As the number of papers published differed by journal and year (Table 1) from a low of 55 (2019, *Molecular Psychiatry*) to a high of 1067 (2009, *Journal of Neuroscience*), we used proportional variables within each analysis. Statistical analyses were performed using Statistica v13. Data were reported and analysed as percentages of total papers per journal per year. We used proportional data to run general linear analysis of variance (ANOVA) across year (2009, 2019) and discipline (neuroscience, psychiatry) with our dependent variables of interest. We also used method of analyses (complete analysis by sex, covariate, main effect, statistics not given, analysed separately, mixed), single-sex studies (male, female) and country of author affiliation (USA, Canada, UK, EU, Asia, Combination) as within-subjects factors. Post-hoc comparisons used Newman–Keuls comparisons. Chi-square analyses were conducted to determine if inferred-sex of author influenced the proportion of papers that used optimal analyses for discovery of sex differences or that used single-sex studies. Significance was set at $\alpha = 0.05$, and all tests were two-tailed unless otherwise noted. Effect sizes using $n_p^2$ or Cohen's $d$ are provided. All analyses were tested for assumptions of ANOVA using Bartlett's test of homogeneity of variance and Kolmogorov–Smirnov test for normality. None of the variables violated assumptions except for male-only papers and these data were transformed prior to analysis.

**Reporting summary.** Further information on research design is available in the Nature Research Reporting Summary linked to this article.

## Data availability
The processed source data (SourceData_All_Journals_2009_2019.tab) used in this study are available in the Dataverse repository, https://doi.org/10.5683/SP3/VDH895. The source data was obtained from publicly available repositories contained in each of the journals sampled websites. Source data are provided with this paper.

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

## Acknowledgements

This research was funded by a Natural Sciences and Engineering Research Council of Canada grant to LAMG (2018-04301) and supported by the BC Women's Foundation (AWD-020262 award to LAMG).

## Author contributions

R.K.R. and T.F.L.S. collated and tabulated the data with consultation with L.A.M.G. and A.A. T.E.H., A.A. and L.A.M.G. carried out the analysis. R.K.R., T.F.L.S. and L.A.M.G. wrote the manuscript. L.A.M.G. supervised the work.

## Competing interests

The authors declare no competing interests.
