## [Peer Review File · Nature Communications]

An analysis of neuroscience and psychiatry papers published from 2009 and 2019 outlines opportunities for increasing discovery of sex differencesREVIEWER COMMENTS

Reviewer #1 (Remarks to the Author):

This manuscript provides another analysis of whether the scientific community is adapting to the new guidelines, mandates or urgings of funding agencies and publishers in North America and parts of Europe to being to incorporate both sexes in neuroscience research and to analyze data for the effects of sex. The strengths of this analysis are the large number of articles examined and the careful scrutiny of both how the experiments were designed and how the results were analyzed. The weaknesses are that they reach essentially the same conclusion as the other recent studies, which they cite, and they provide no clear assessment as to why things have not improved, fail to identify some additional sources of potential lack of compliance and do not provide any suggestions for new guidelines or initiatives beyond that we should all just do better.

Some specific suggestions below.

1) There is a lack of coherent presentation of the time-line of events regarding SABV, SGBA and SAGER and the current analysis. One very useful figure would be to present such a graphic, including the country of origin of each of the initiatives. What would become immediately evident is that the current analysis is not of the impact of 10 – years of i of new mandates etc (i.e. SABV), but rather only, at the most, 3 years since the NIH did not announce the mandate for SABV until 2016. SGBA was deployed in 2019 and SAGER was not until 2020. Thus there is a disconnect between the efforts to improve representation of both sexes in neuroscience research and the current analysis. This needs to be clearly acknowledged as a weakness.

2) And pursuant to the above, the title should be changed to reflect such.

3) Likewise on pg 13 the authors state – “Our exhaustive survey of 3191 papers across six journals in Neuroscience and Psychiatry revealed some interesting insights into the adoption of SABV over the ten year period from 2009 to 2019.” The term SABV did not enter the lexicon until 2016.

1) The authors seem to take a very chiding view that only a small percentage of studies use sex as a primary discovery variable. However, it was never the mandate of SABV that researchers study sex differences, indeed this was explicitly avoided due to the clear financial and other consequences. The authors have an excellent point that using sex as a co-variate may be obscuring some findings on the influence of sex, but as they word it, it might not be clear to the reader precisely the point they are trying to make, at least it was not clear to this reviewer. It should be very clear that not every experiment by every researcher needs to include equal numbers of males and females. Better would be to cite studies which have modeled how well balanced the sex ratio needs to be and the sample size to detect a sex difference of a particular effect size, such as that by Beery (Curr Opinion in Behav Sci – Figure 2). It seems highly likely that the 6% number of studies they frequently mention as appropriately assessing for the impact of sex were studies in which that was the entire goal. This is a very different thing from the overwhelming (~95%?) of studies in neuroscience and psychiatry which have a separate and equally valid goal. Yes they should consider sex as variable that influences the findings, but they do not need to design their studies to study a sex difference.

2) When comparing the shift over the 10 year period for neuroscience versus psychiatry, the authors should acknowledge that psychiatry was already well ahead of neuroscience. As they note, this is likely because of the higher representation of human subjects, but this is also at least partly likely due to the 1993 act of congress in the US mandating equal representation of women, children and minorities in all clinical trials. In this case there has been a much longer period of time for the mandate to take effect, and that does seem to be evident here.

3) The presentation of the findings is framed entirely around initiatives regarding inclusion of both sexes in research that originated in the US, Canada and parts of Europe, each of which has varying levels of “mandateness” and only one of which actually speaks to the policies of journals (SAGER), which is both very recent and only a suggestion. Yet the data generated here rely entirely on the published literature which has no rules or restrictions on the use of both sexes and which include scientists from around the world. The degree to which country of origin is diluting any improvements in the incorporation of SABV is entirely unknown and is unfortunately a missed opportunity from this analysis. It would be of considerable interest to know if scientists from the US, and possibly Canada,

were more likely to conduct a factor analysis using sex as a variable than those from say China, or Germany. It would certainly not be fair to ask the authors to re-analyze for the effect of country of origin but at the least this major short-coming needs to be acknowledged and at best, some data on the relative percent of publications from various global regions would be a welcome and informative inclusion. As is, the data are somewhat analogous to trying to understand the impact of speed limits in the US on drivers all over the world.

4) Lastly, while overall the manuscript is well written, it is at times quite repetitive and the precise presentation of each % change makes for heavy sledding. Use of more descriptive terms in the text and leaving the numbers to the figures would be helpful.

Reviewer #2 (Remarks to the Author):

This manuscript from Rechlin and colleagues provides a critical examination of the state of sex omission and bias in the neuroscience and neuropsychological research. This manuscript is a highly valuable update to previous analyses of this literature. This manuscript also provides novel insight by providing a comprehensive analysis of how sex is being used as an experimental variable, as well as the various justifications given for the choices made by research scientists. Specifically, the real value of this manuscript lies in that the authors analyzed how males and females were included in data analyses. Unlike previous examinations, the authors determined not only whether a statistical analysis of sex as a biological variable was performed, but how the analysis was performed. The authors also performed a novel assessment of whether best practices were performed. Overall, the significance will be high, the work supports the conclusions and claims, there are no critical flaws in the data analysis or conclusions, and the methodology sound. Enough detail is provided to reproduce the study. I have a few minor comments to improve this manuscript. Overall this manuscript is important to publish.

1. Methods. Please provide the definition of cell lines as used in this study. Did cell lines include only immortalized cells, or also primary cell cultures? What about the various stem-cell derivatives?

2. Methods. Were manuscripts with an inconsistent sex-based analysis excluded from the study? If not, please define how results from these manuscripts were incorporated into the study.

3. Methods. How were manuscripts which included multiple research models handled? Were they excluded or included from the study? If the manuscripts were included what was the data analysis protocol employed?

4. Results. The number of papers in each specific group as well as in the statistical comparisons is difficult for me to ascertain. One option to address this omission would be to include a new comprehensive table documenting the experimental n of all groups. However this omission could also be potentially fixed either via textual edits, and/or additions of experimental n designation in the figures or augmentation of existing tables. The bottom line is that the final "experimental n" per group requires better documentation.

Reviewer #3 (Remarks to the Author):

This meta-analysis examines the practice of including both sexes in neurobiology and psychiatry research papers. The authors used three high profile journals in each field, sampling papers from 2009 and 2019 to determine whether initiatives such as SABV have had an impact. The authors conclude that although the proportion of papers that include both sexes has increased across the 10 year span, male-only papers still vastly outnumber female-only papers, and most "both sexes" papers do not go far enough in formally assessing the influence of sex on experimental outcomes. The paper is very well written and I think will be a useful tool for neuroscientists in a broad range of

subdisciplines. I have a few points that I think will improve clarity and utility.

1. The word "optimal," which is used throughout the manuscript, does not sit well with me without demonstration of what the "optimal" approach (if there is one) is. I would recommend tempering this language or justifying the use of the word better within the manuscript.
2. The authors distinguish between six different kinds of analyses that a broad readership (such as that for Nature Comms) may not be familiar with. I think these need to be defined and explained within the context of SABV.
3. I really like the infographic upside-down pyramid at the beginning, but I would have thought that "both sexes," "male only," and "female only" layers should add up to 100%, yet they only add up to 83%. If the remaining 17% are papers in which the sex of the subjects was not stated, I think this should be an included layer.
4. In Figure 5B, I think there may be a typo in the labels - since all papers used both sexes, should the categories be "male-skewed" etc instead of "male only"?

Reviewer #1

This manuscript provides another analysis of whether the scientific community is adapting to the new guidelines, mandates or urgings of funding agencies and publishers in North America and parts of Europe to being to incorporate both sexes in neuroscience research and to analyze data for the effects of sex. The strengths of this analysis are the large number of articles examined and the careful scrutiny of both how the experiments were designed and how the results were analyzed. The weaknesses are that they reach essentially the same conclusion as the other recent studies, which they cite, and they provide no clear assessment as to why things have not improved, fail to identify some additional sources of potential lack of compliance and do not provide any suggestions for new guidelines or initiatives beyond that we should all just do better.

We now include a section at the end of the discussion with potential reasons for lack of compliance, and have suggestions for researchers, publishers and funders in a new section in the Discussion called “**Call to Action: Fixing Implementation Issues with Carrots and Sticks?**” We have also included more data that help inform these suggestions.

1) There is a lack of coherent presentation of the time-line of events regarding SABV, SGBA and SAGER and the current analysis. One very useful figure would be to present such a graphic, including the country of origin of each of the initiatives. What would become immediately evident is that the current analysis is not of the impact of 10 – years of i of new mandates etc (i.e. SABV), but rather only, at the most, 3 years since the NIH did not announce the mandate for SABV until 2016. SGBA was deployed in 2019 and SAGER was not until 2020. Thus there is a disconnect between the efforts to improve representation of both sexes in neuroscience research and the current analysis. This needs to be clearly acknowledged as a weakness.

We thank the reviewer for the idea to include a figure to show a timeline of inclusion of the use of males and females in biomedical and clinical research from the European Union (EU), Canada and US. As the reviewer will see, the timeline (Supplemental Figure 1) makes it clear that many of these initiatives have been around much earlier in Canada and Europe, although NIH was the first to act on clinical trials (1993) and in clinical research (2001). The point on mandates is well taken, but our data indicating that the majority of papers (68% in 2019) now include both male and females suggest that the time is ripe to do a deep dive on these studies, and that the ten-year span is appropriate. However, we agree with the point on the newer mandates and the lexicon (point 3 below), and we have reflected on this in the manuscript in the Discussion.

2) And pursuant to the above, the title should be changed to reflect such.

We did not change the title as we are not referring to mandates in the title.

3) Likewise on pg 13 the authors state – “Our exhaustive survey of 3191 papers across six journals in Neuroscience and Psychiatry revealed some interesting insights into the adoption of SABV over the ten year period from 2009 to 2019.” The term SABV did not enter the lexicon until 2016.

We have altered our wording.

1) The authors seem to take a very chiding view that only a small percentage of studies use sex as a primary discovery variable. However, it was never the mandate of SABV that researchers study sex differences, indeed this was explicitly avoided due to the clear financial and other consequences. The authors have an excellent point that using sex as a co-variate may be obscuring some findings on the influence of sex, but as they word it, it might not be clear to the reader precisely the point they are trying to make, at least it was not clear to this reviewer. It should be very clear that not every experiment by every researcher needs to include equal numbers of males and females. Better would be to cite studies which have modeled how well balanced the sex ratio needs to be and the sample size to detect a sex difference of a particular effect size, such as that by Beery (Curr Opinion in Behav Sci – Figure 2). It seems highly likely that the 6% number of studies they frequently mention as appropriately assessing for the impact of sex were studies in which that was the entire goal. This is a very different thing from the overwhelming (~95%?) of studies in neuroscience and psychiatry which have a separate and equally valid goal. Yes they should consider sex as variable that influences the findings, but they do not need to design their studies to study a sex difference.

We apologise if we sound chiding. We are excited that people are including males and females and getting people to understand that studying them may prove fruitful, and even clean up their variability. However, it is also important that beyond the inclusion of the use of both sexes, that the researchers examine the influence of sex in their work. Otherwise what would be the point? We have tried to make this clearer throughout the document. The issue of not designing studies to study a sex difference is certainly important and we now acknowledge this. However, in our survey we found that even in studies that used both males and females that 75% of these were not disclosing sample size, not using both sexes consistently across studies in the paper, and finally using unequal proportions of sexes. This can lead to a reduction in power and has implications for analyses. Indeed, some of the unbalanced samples size examples are extreme - one 2019 *Nature Neuroscience* study used 10 female, 2 male in the controls, and 7 females and 21 males in the treatment(transgenic) condition. It is this type of disproportionate use of sexes across groups that will undoubtedly lead to reduced power and increased variability (and it's almost impossible to understand if there is a sex difference under those conditions). We now include more information from Beery and others (Busch et al., 2019; Galea et al., 2020) on how power is affected by sample size when examining males and females. If there is a strong interaction or main effect present then modelling indicates power is increased. However, it should also be noted if the sex effect is not 50% in one sex from the other larger samples sizes are needed (Galea et al., 2020).

We have expanded sections on how a covariate obscures findings of possible sex differences in research. Indeed, not every experiment needs to employ equal numbers of sexes and there are very good reasons for male-only and female-only studies. We do not dispute that (we do point out that there are 9x fewer studies using female-only which causes another disparity). Our goal in providing these data are not to ensure everyone is doing sex difference research but to show that the method and analyses employed by the vast majority of studies will not lead to fruitful investigations and that analysing with sex as a discovery variable may lead to improved power if there are large sex differences present.

2) When comparing the shift over the 10-year period for neuroscience versus psychiatry, the authors should acknowledge that psychiatry was already well ahead of neuroscience. As they note, this is likely because of the higher representation of human subjects, but this is also at least partly likely due to the 1993 act of congress in the US mandating equal representation of women, children and minorities in all clinical trials. In this case there has been a much longer period of time for the mandate to take effect, and that does seem to be evident here.

We gratefully acknowledge this comment and have included this in our discussion. Indeed, this explanation does fit our data as there was only a 10% increase in the proportion of studies in Psychiatry using both sexes across ten years but a 70% increase in the Neuroscience discipline. The mandates are working effectively here to increase use of both sexes. Also, it's important to note that the US contributes just less than half of the studies overall. The mandate by NIH is important but it is not the only funder of research and it's important to recognise this.

3) The presentation of the findings is framed entirely around initiatives regarding inclusion of both sexes in research that originated in the US, Canada and parts of Europe, each of which has varying levels of "mandateness" and only one of which actually speaks to the policies of journals (SAGER), which is both very recent and only a suggestion. Yet the data generated here rely entirely on the published literature which has no rules or restrictions on the use of both sexes and which include scientists from around the world. The degree to which country of origin is diluting any improvements in the incorporation of SABV is entirely unknown and is unfortunately a missed opportunity from this analysis. It would be of considerable interest to know if scientists from the US, and possibly Canada, were more likely to conduct a factor analysis using sex as a variable than those from say China, or Germany. It would certainly not be fair to ask the authors to re-analyze for the effect of country of origin but at the least this major short-coming needs to be acknowledged and at best, some data on the relative percent of publications from various global regions would be a welcome and informative inclusion. As is, the data are somewhat analogous to trying to understand the impact of speed limits in the US on drivers all over the world.

We agree and include these data in Figure 5. We have documented and analysed the countries of origin for total papers and optimal analyses for discovery of possible sex differences (sex as a discovery variable). As noted in the paper the proportion of papers that use sex as discovery variable does increase for papers that originated in the US, Canada, EU or a combination of countries. We discuss these data on page 15 and 20.

4) Lastly, while overall the manuscript is well written, it is at times quite repetitive and the precise presentation of each % change makes for heavy sledding. Use of more descriptive terms in the text and leaving the numbers to the figures would be helpful.

We have cut down on repetitive sentences and the use of the precise percentage in the main text.

Reviewer #2 (Remarks to the Author):

This manuscript from Rechlin and colleagues provides a critical examination of the state of sex omission and bias in the neuroscience and neuropsychological research. This manuscript is a highly valuable update to previous analyses of this literature. This manuscript also provides novel insight by providing a comprehensive analysis of how sex is being used as an experimental variable, as well as the various justifications given for the choices made by research scientists. Specifically, the real value of this manuscript lies in that the authors analyzed how males and females were included in data analyses. Unlike previous examinations, the authors determined not only whether a statistical analysis of sex as a biological variable was performed, but how the analysis was performed. The authors also performed a novel assessment of whether best practices were performed. Overall, the significance will be high, the work supports the conclusions and claims, there are no critical flaws in the data analysis or conclusions, and the methodology sound. Enough detail is provided to reproduce the study. I have a few minor comments to improve this manuscript. Overall this manuscript is important to publish.

Thank you for your positive comments.

1. Methods. Please provide the definition of cell lines as used in this study. Did cell lines include only immortalized cells, or also primary cell cultures? What about the various stem-cell derivatives?

All cell lines were included in this study. We now provide a percentage of the cells that were used. We relied on the authors to provide information about sex of cells, even when sex of cells is known with some immortalised cell lines. These data are now included in Figure 2c

2. Methods. Were manuscripts with an inconsistent sex-based analysis excluded from the study? If not, please define how results from these manuscripts were incorporated into the study.

Manuscripts with inconsistent analyses were included. They were categorized as “mixed analysis”. Mixed analysis included any papers which used some form of sex-based analysis their experiment but were not consistent throughout the study. This is detailed on pages 6-7.

3. Methods. How were manuscripts which included multiple research models handled? Were they excluded or included from the study? If the manuscripts were included what was the data analysis protocol employed?

Manuscripts which included multiple research models were included. In Figure 2 which represents the breakdown of papers by research model, manuscripts which included multiple models would be represented more than once. For example, some studies included mice and rats, this would have increased the frequency of the mice and rat category by one. All analyses were done proportionate to the total number of papers in the journal and year (not to the model systems).

4. Results. The number of papers in each specific group as well as in the statistical comparisons is difficult for me to ascertain. One option to address this omission would be to include a new comprehensive table documenting the experimental n of all groups. However

this omission could also be potentially fixed either via textual edits, and/or additions of experimental n designation in the figures or augmentation of existing tables. The bottom line is that the final "experimental n" per group requires better documentation.

Thank you for this suggestion. We now have added the n to each figure and table and in the manuscript where appropriate.

Reviewer #3 (Remarks to the Author):

This meta-analysis examines the practice of including both sexes in neurobiology and psychiatry research papers. The authors used three high profile journals in each field, sampling papers from 2009 and 2019 to determine whether initiatives such as SABV have had an impact. The authors conclude that although the proportion of papers that include both sexes has increased across the 10 year span, male-only papers still vastly outnumber female-only papers, and most "both sexes" papers do not go far enough in formally assessing the influence of sex on experimental outcomes. The paper is very well written and I think will be a useful tool for neuroscientists in a broad range of subdisciplines. I have a few points that I think will improve clarity and utility.

Many thanks for your comments.

1. The word "optimal," which is used throughout the manuscript, does not sit well with me without demonstration of what the "optimal" approach (if there is one) is. I would recommend tempering this language or justifying the use of the word better within the manuscript.

We have defined what we mean by "optimal," and now consistently use the phrase "optimal for discovery of sex differences." For design we refer to using a relatively equal proportion of sexes, disclosing sample size and use of both sexes throughout the paper. These design issues accounted for 75% of the time when both males and females were used in papers. If the paper does not acknowledge how many males or females were used this is suboptimal for the reader. If the researchers did not use both sexes in all experiments and if relatively equal numbers of males and females were not used this has implications for variability/power in the statistical analyses and will hamper discovery of sex differences. We explain this further on page 7.

2. The authors distinguish between six different kinds of analyses that a broad readership (such as that for Nature Comms) may not be familiar with. I think these need to be defined and explained within the context of SABV.

These are described in more detail on pages 6-7.

3. I really like the infographic upside-down pyramid at the beginning, but I would have thought that "both sexes," "male only," and "female only" layers should add up to 100%, yet they only add up to 83%. If the remaining 17% are papers in which the sex of the subjects was not stated, I think this should be an included layer.

Thank you for noticing this omission of sex omission - this has been included and unfortunately Nature Communications does not accept graphical abstracts so we include this now as Figure 6 in Discussion (we included a different figure showing the differences between 2009 and 2019).

4. In Figure 5B, I think there may be a typo in the labels - since all papers used both sexes, should the categories be "male-skewed" etc instead of "male only"?

Thank you for noticing this - it has been corrected.

REVIEWER COMMENTS

Reviewer #1 (Remarks to the Author):

The manuscript is improved and while I might still argue with some of the authors interpretation of the data, it is in the end their analysis and therefore their prerogative to frame it as they wish

A few typo's and minor comments

1) Pg 4 – line 78 – into the scoring of “grants”

2) Pg 5 – line 132 – typo “discovery of possible sexes”

3) Pg 10 – line 290 – “The percentage of papers failing to disclose sex fell dramatically over the years, with the greatest 291 change seen in Neuroscience as only 3% of papers omitted sex in 2019” – statements like this seem to contradict the title which suggests there has been no change in 10 years

4) Pg 14 – line 424 – the term “North American papers” gives citizenship to an inanimate object. I suggest something like “Manuscripts originating from laboratories in North America...”

5) Pg 20 – the authors propose money dedicated to SABV research is a productive means to increase the use of both sexes in preclinical research. Many NIH institutes and the Office of Research on Women's Health at NIH have partnered to provide large supplements to existing grants that incorporate the opposite sex. This has been going on for at least a decade and so perhaps one could argue in this case it hasn't worked very well?

6) Lastly, the authors are strongly encouraged to cite a similar but also distinct analysis conducted on the same topic and published in eLife this year by Donna Maney (PMID: 34726154).

Reviewer #2 (Remarks to the Author):

The authors have satisfactorily addressed my comments.

Reviewer #3 (Remarks to the Author):

The authors have done a nice job addressing reviewer comments. I just have a couple of minor remaining issues (no re-review necessary, these edits can be assessed editorially):

1. The font in the figures is sometimes exceptionally and unnecessarily small, especially figures 2,3, and 4 (5 to a lesser extent but there is still clearly plenty of white space to work with). Please increase the size substantially so it is more easily read without having to zoom.

2. The pyramid in Figure 6 has added the "sex omitted" segment but now the "optimal design" segment makes it look like everything adds up to over 100%. Presumably, the "optimal design" studies are a sub-division of the "both sexes" group, and so this should be conveyed appropriately in the figure.

Reviewer 1 (Remarks to the Author):

1) Pg 4 – line 78 – into the scoring of “grants”

Fixed

2) Pg 5 – line 132 – typo “discovery of possible sexes”

Fixed

3) Pg 10 – line 290 – “The percentage of papers failing to disclose sex fell dramatically over the years, with the greatest 291 change seen in Neuroscience as only 3% of papers omitted sex in 2019” – statements like this seem to contradict the title which suggests there has been no change in 10 years

We changed the title.

4) Pg 14 – line 424 – the term “North American papers” gives citizenship to an inanimate object. I suggest something like “Manuscripts originating from laboratories in North America...”

We altered our wording.

5) Pg 20 – the authors propose money dedicated to SABV research is a productive means to increase the use of both sexes in preclinical research. Many NIH institutes and the Office of Research on Women’s Health at NIH have partnered to provide large supplements to existing grants that incorporate the opposite sex. This has been going on for at least a decade and so perhaps one could argue in this case it hasn’t worked very well?

This is an important point and has been added to the discussion. In our view this is perhaps due to this funding being a supplement rather than a call for a specific granting mechanism or RFA dedicated to sex differences research. But it is important to assess these initiatives in the future to determine what has worked and what has not.

6) Lastly, the authors are strongly encouraged to cite a similar but also distinct analysis conducted on the same topic and published in eLife this year by Donna Maney (PMID: 34726154).

Thank you for pointing out this paper, which we had seen but unfortunately came out after we had resubmitted our paper. We now refer to it in the introduction and discussion.

Reviewer #2 (Remarks to the Author):

The authors have satisfactorily addressed my comments.

Thank you.

Reviewer #3 (Remarks to the Author):

The authors have done a nice job addressing reviewer comments. I just have a couple of minor remaining issues (no re-review necessary, these edits can be assessed editorially):

Thank you.

1. The font in the figures is sometimes exceptionally and unnecessarily small, especially figures 2,3, and 4 (5 to a lesser extent but there is still clearly plenty of white space to work with). Please increase the size substantially so it is more easily read without having to zoom.

We used the formatting required by the journal. But we are happy to change this if allowed.

2. The pyramid in Figure 6 has added the "sex omitted" segment but now the "optimal design" segment makes it look like everything adds up to over 100%. Presumably, the "optimal design" studies are a sub-division of the "both sexes" group, and so this should be conveyed appropriately in the figure.

Figure 6 has been edited for clarity (the optimal design and optimal analyses are have a different pattern).